On the influence of artificially distorted images in firearm detection performance using deep learning

Corral-Sanz Patricia
Barreiro-Garrido Alvaro
Moreno A. Belen
Sanchez Angel angel.sanchez@urjc.es
Department of Computer Science and Statistics, Universidad Rey Juan Carlos , Mostoles , Madrid , Spain
Angiulli Giovanni
Electronic publication date: 2024 Oct 18
Publication date: 2024
Volume: 10
Electronic Location ID: e2381
Received 2024 May 16; Accepted 2024 Sep 10
Copyright: ©2024 Corral-Sanz et al.
Copyright year: 2024
Copyright holder: Corral-Sanz et al.
License: This is an open access article distributed under the terms of the Creative Commons Attribution License, which permits unrestricted use, distribution, reproduction and adaptation in any medium and for any purpose provided that it is properly attributed. For attribution, the original author(s), title, publication source (PeerJ Computer Science) and either DOI or URL of the article must be cited.
License URL: https://creativecommons.org/licenses/by/4.0/

Keywords: Artificial image distortions, Firearm detection, Deep learning, YOLO, Object detection metrics

Funding: State R+D+i Programme, Spanish Ministry of Science, Innovation and Universities PID2021-124064OB-I00 This research was supported by State R+D+i Programme, Spanish Ministry of Science, Innovation and Universities, Grant no. PID2021-124064OB-I00. The funders had no role in study design, data collection and analysis, decision to publish, or preparation of the manuscript.

==============================
Detecting people carrying firearms in outdoor or indoor scenes usually identifies (or avoids) potentially dangerous situations. Nevertheless, the automatic detection of these weapons can be greatly affected by the scene conditions. Commonly, in real scenes these firearms can be seen from different perspectives. They also may have different real and apparent sizes. Moreover, the images containing these targets are usually cluttered, and firearms can appear as partially occluded. It is also common that the images can be affected by several types of distortions such as impulse noise, image darkening or blurring. All these perceived variabilities could significantly degrade the accuracy of firearm detection. Current deep detection networks offer good classification accuracy, with high efficiency and under constrained computational resources. However, the influence of practical conditions in which the objects are to be detected has not sufficiently been analyzed. Our article describes an experimental study on how a set of selected image distortions quantitatively degrade the detection performance on test images when the detection networks have only been trained with images that do not present the alterations. The analyzed test image distortions include impulse noise, blurring (or defocus), image darkening, image shrinking and occlusions. In order to quantify the impact of each individual distortion on the firearm detection problem, we have used a standard YOLOv5 network. Our experimental results have shown that the increased addition of impulse salt-and-pepper noise is by far the distortion that affects the most the performance of the detection network.

Introduction

Weapon detection problem consists in locating and classifying in different environments the potential threats caused by the presence of firearms, explosives, knives and other dangerous objects (Paulter, 2015; Xu, 2021). This task can use a variety of sensing methods such as X-ray scanning, metal detection, and specially images/videos (e.g., visible or thermal), which in many cases come from Closed Circuit Television (CCTV) systems, in order to detect and identify those threats. Therefore, weapon detection technology helps to prevent violent incidents and ensures the safety of individuals specially in different urban environments (Debnath & Bhowmik, 2021). For example, in places like stadiums, government buildings, airports or schools, where large numbers of people gather, the risk of a violent incident happening is higher. This technology also helps to discourage individuals from attempting to bring weapons into these places. Overall, weapon detection is an important tool for keeping safety in high-risk environments.

A significant part of image-based firearm detection technology is represented by the video surveillance CCTV cameras (Gelana & Yadav, 2019). These cameras are being installed in many public outdoor places to provide security. In these environments, the image capture conditions are not controlled at all and images could present distortions such as: impulse noise due to poor illumination conditions, blurring due to camera defocus or movements of objects, occlusions due to the variegated complexity of the scenes, and object shrinking due to long distances between camera and objects, among others. All these mentioned distortions hinder to varying degrees the corresponding object detection tasks. In this context, the present study, applied to firearm detection, aims to quantify the influence of different perceived types of image distortions in the task of detecting these objects accurately. In our study, the analyzed image degradations correspond to some common distortions appearing in the context of realistic firearm images, such as occlusions, including noise, blurring, darkening and shrinking the images that, at certain levels, could significantly degrade the detection performance. To quantify the specific impact of these image conditions, we produced altered test images from the original ones by adding to them the corresponding distortion with different levels or “strengths” (e.g., impulse noise at 2%). Note that the training images do not include the aforementioned distortions.

Considered distortions are intended to simulate some common realistic conditions that hinder firearm detection in scenes. In particular, testing image detection models with the considered distortions is crucial for ensuring their robustness, reliability, and generalization ability in real-world applications. A use of the present study could be an assessment method to help to interpret the correctness and accurateness of firearm detections in test images, after analyzing each distortion type (and its “strength”) contained in a given image. An advantage of using artificial transformations on images is that we can analyze the influence of the image distortion degree, which is a much more complex task when dealing with natural distorted images (i.e., “in-capture” distortions).

We conduct a set of comprehensible experiments to quantitatively analyze the influence of the considered individual image distortion factors in the results of the firearm detection task itself. For our purpose, we have chosen a well-established one-stage object detection system, as it is the case for YOLOv5 (Jocher, 2020). This detector is lightweight, effective, it offers good inference times and it is also capable of achieving very good results for object detection tasks. Although new versions of YOLO detectors have recently appeared (e.g., YOLOv8 (Terven & Cordova-Esparza, 2023)), in our experiments we have used YOLOv5 since it is easier to train and it is also a suitable choice when one needs to deploy a solution on devices without GPU support.

Related work

This subsection outlines some relevant works on firearm detection using deep learning, as well as some studies concerning perceived image distortions.

Firearm detection using deep learning

The problem of detecting weapons in images has been intensively researched in the last decades (Darker, Gale & Blechko, 2008; Bhatti et al., 2021). First, through traditional machine learning methods and more recently using deep learning models, specially convolutional neural networks (Yadav, Gupta & Sharma, 2023).

Most research on weapon detection has been targeted toward knives (Kmiec & Glowacz, 2011; Buckchash & Raman, 2017; Castillo et al., 2019; Glowacz, Kmieć & Dziech, 2015) and firearms (Tiwari & Verma, 2015; Olmos, Tabik & Herrera, 2018; Bhatti et al., 2021). Detection of knives and firearms in images has received a growing attention by the scientific community due to their security implications (Debnath & Bhowmik, 2021). Some works are focused on the detection of only one of these types of weapons (e.g., Moran, Conci & Sanchez, 2022) only considers the detection of knives and (Olmos, Tabik & Herrera, 2018) is solely devoted to handgun firearms, while other consider both detections jointly (Grega et al., 2016). Weapon detection is generally a challenging task due to variable size and shapes of objects within the images, possible occlusions and/or cluttered backgrounds of scenes. A related problem is to detect paired bounding boxes that contain both the weapon and the human, for a more robust visual identification of gunmen in crowds (Basit et al., 2020; Mahmood et al., 2024).

Nowadays, deep learning has revolutionized the general object detection problem (Zou et al., 2019). In particular, deep learning firearm detection has also been recently investigated. Lai & Maples (2017) used CNN for detecting and classifying weapons in images. These authors considered around 3,000 CCTV images with weapons as training data to cover every situation and possible orientation of these objects. Olmos, Tabik & Herrera (2018) proposed a CNN-based system for detecting handguns from CCTV videos with the goal of reducing false positives. These authors created their own dataset (called DaSCI), and achieved their best results with a Faster R-CNN model. Salido et al. (2021) compared the performance of three deep CNN models: Faster R-CNN, RetinaNet and YOLOv3, respectively, in the detection of guns in videos. The study focuses on how the inclusion of gun grip information influences the results of each detector. Only a consistent improvement is achieved using YOLOv3. Khan et al. (2023) used U-Net networks for weapon segmentation in real time when people pass through a scanner system. For this purpose, the 2D segmentation network is reformulated and a Gaussian map is used to model the weapons in the feature map. Their dataset only provides handguns such as pistols or revolvers. Ruiz-Santaquiteria et al. (2023) recently proposed a combination of a human body pose classifier (OpenPose) with a deep network that processes images to extract relevant features for gun detection in video surveillance. After comparing the results obtained with different deep networks (ResNet-50, EfficientNet-B4, ConvNeXt-Base, Darknet53, DeiT and ViT, respectively) in combination with and without the pose features and filtering out false positives, the authors concluded that the best detection results were obtained with the Vision Transformer (ViT) model. The detection of firearms orientation can provide insights about the behavior and intentions of people carrying these weapons, which is critical for identifying potential threats. Iqbal et al. (2021) propose a weakly supervised deep learning architecture to predict Oriented Bounding Boxes (OBB) without using OBB annotations while training.

Perceived image distortions

We have not found in the literature any experimental studies aimed at analyzing separately the influence of synthetic image distortions (i.e., by applying artificially computed blur, impulse noise and random occlusions) on specific object detection problems with the approach presented here. For a particular detection problem, our approach consists in using a set of training images without the considered distortions and, after that, using new distorted test images to quantify the impact that the distortions and their “degrees” have on the accuracy of the detection task.

The work by Dodge & Karam (2016) used a classification problem to understand how image quality affects different deep neural networks. These authors trained their networks with distorted images using different levels of quality distortions. Then, they carried out a study on the effect of compression, noise blur and contrast.

There are also some studies that, using deep networks, analyze only the impact of other isolated image variabilities. For instance, the effect that illumination conditions (Sanchez et al., 2016) have on some specific object detection problems such as face recognition. Other authors consider the scale of the YOLOv5 model for blood cell detection (Rahaman et al., 2022) or its application for classification of caries lesions (Salahin et al., 2023). The work by Venkataramanan et al. (2022) analyzed the detection problem in authentically distorted images of roadways for quality assessment of detection algorithms.

Contribution and outline of this work

This article describes an experimental study that analyzes individually how each considered image distortion quantitatively affects the detection performance on test images when the detector network has not been trained with images presenting these variations. The main contribution of this work consists in quantifying, analyzing and comparing the effect of each considered individual image distortion (e.g., occlusion, impulse noise or blurring) in the detection performance. It should also be noted that that the goal of this study is not demonstrating the viability of the considered YOLO model itself for our firearm detection problem, but quantifying how it gets affected by the aforementioned distortions.

The rest of the article is organized as follows: Section ‘Materials and Methods’ introduces the materials and methods used in this research on detection of firearms. Section ‘Results’ describes the experimental setup, and it also displays and analyzes the results achieved for the different experiments on the considered firearm detection problem. Section ‘Discussion’ discusses these results and points out the most relevant findings. Finally, in Section ‘Conclusions’ we summarize the conclusions of the present work.

Materials and Methods

This section first describes the object detection problem in images and the neural detector model used in the experiments. Then, an overview of the stages in the proposed solution is presented. The preprocessing and data augmentation applied to the original training images is later indicated. We continue with the value of hyperparameters of the YOLO detector employed, and we also provide some details on the training of the network. Finally, the dataset used for the experiments is described.

Object detection and the YOLO model

Object detection is a challenging task in Computer Vision that has received large attention in recent years, especially with the development of deep learning (Zou et al., 2019; Wang et al., 2021). It presents many applications related to video surveillance, automated vehicle system, robot vision or machine inspection, among many others. The problem consists in recognizing and localizing some classes of objects present in static images or videos.

Recognizing (or classifying) involves identifying the categories of all object instances in a scene from a given set of classes, along with their confidence values. Localizing, in contrast, returns the coordinates of bounding boxes for each detected object in the image. Detection differs from instance segmentation, which identifies the object instance each pixel belongs to. Challenges in object detection include geometrical variations (e.g., scale changes, small object-to-image size ratios), partial occlusions, or varying illumination. Some images may exhibit multiple variabilities, such as small and partially occluded objects.

The You Only Look Once (YOLO) model, proposed by Redmon et al. (2016), is a state-of-the-art real-time object detection network. YOLO is a one-stage detector that uses features from the entire image to predict class probabilities and bounding box coordinates in a single pass. It formulates object detection as a regression problem, enhancing speed, accuracy, and generalization. YOLO splits an image into an NxN grid, where each cell predicts the presence of one object using a fixed number of bounding boxes and a Non-Maxima Suppression (NMS) algorithm. The YOLO framework has evolved through iterations like YOLOv8, YOLO-NAS, and YOLO with Transformers. We focused on YOLOv5 for this study, developed by Ultralytics in 2020 using Python and PyTorch, offering versions from nano to extra large to suit various hardware requirements. We tested YOLOv5s (small) and YOLOv5m (medium) configurations.

Figure 1 depicts the simplified network architecture of YOLOv5, comprising three main components: backbone, neck, and head. A 416 × 416 RGB image is processed through an input layer to the backbone, a modified CSP Darknet53 CNN, which extracts hierarchical features at various scales using the Cross Stage Partial (CSP) strategy, enhancing inference speed by reducing parameters. The neck integrates output features from the backbone at different resolutions using modules like Fast Spatial Pyramid Pooling (FSPP) and Path Aggregation Networks (PAN), connecting the backbone to the head. The head, based on anchors, classifies detected objects with three convolutional layers, predicting bounding box locations, confidence scores, and classes, displaying this information in the output image.

Figure 1 Schematic layer representation of YOLOv5 architecture.

(Note: The image within the figure can be found in the DaSCI Weapon Detection dataset, and it is distributed under the terms of the Creative Commons Attribution 4.0 International).

It is worth mentioning that this YOLO model uses data augmentation during each training batch. The data loader makes three types of augmentations: scaling, colors space adjustments and mosaic (i.e., a combination of four images into four tiles of random ratio), respectively.

Experimental setup

Figure 2 shows a UML diagram with the steps followed in the proposed experimental setup using the considered YOLO model. First, the original set of training images was preprocessed and augmented to increase both the size and the variability of the dataset. These two stages will be described in detail during the next subsection. After that, the chosen YOLO model was trained, tested and evaluated through different experiments using standard metrics for the object detection problem.

Figure 2 Overview of proposed experimental setup for firearms detection.

(Note: The image within the figure can be found in the DaSCI Weapon Detection dataset, and it is distributed under the terms of the Creative Commons Attribution 4.0 International).

Image preprocessing and data augmentation

The original set of training images was first preprocessed and then augmented to increase the quality and size of the original training dataset. The Roboflow tool (Dwyer & Hansen, 2022) was used for all preprocessing tasks. These tasks consisted in first applying a contrast stretching by means of an adaptive equalization (using the Auto-Adjust Contrast command) on the training images. Then, the resulting images were rescaled (using the Resize+Fit (Black Edges) command) from their original dimensions to the YOLOv5 input layer dimension (416 × 416). This rescaling keeps the aspect ratio of source images, and in some cases it creates a black padding image region.

For the augmentation on the training set of images, we used the image Augmentor software (Bloice, Stocker & Holzinger, 2017). This generated five new images for each original one in the dataset according to the following transformations:

1. 45° clockwise rotation (Rotate45 command) followed by horizontal mirroring (flip-left–right command)

2. vertical mirroring (flip-top-bottom command)

3. 25° clockwise rotation (Rotate25 command)

4. 90° clockwise rotation (Rotate90 command) followed by a translation of 40 and 20 pixels respectively in x and y axes (translation-xy(40, 20) command), and

5. vertical mirroring (flip-top-bottom command) followed by 45° clockwise rotation (Rotate45 command)

Figure 3 shows, from top to bottom and from left to right, the application of the five aforementioned augmentations for an original sample training image (upper left corner).

Figure 3 Sample firearm image (upper left corner) and its five augmented images.

(Note: The original image within this figure comes from Wikimedia Commons, and it is distributed under the terms of the Creative Commons Attribution-Share Alike 3.0 Unported license).

The choice of these preprocessing and augmentation transformations was made after multiple experiments.

Network parameterization and training details

Training YOLO models need from a large collection of input images as well as their corresponding output ones with the ground truth boxes for each object instance contained in them. In our approach, we have used transfer learning, and the pretrained weights of a YOLOv5 network trained on the Microsoft (MS) COCO dataset (Lin et al., 2014) were used to boost the training of the model with our images. The MS COCO dataset includes objects belonging to 80 different classes, but unfortunately, some classes like ‘firearm’ or ‘handgun’ are not included. Nonetheless, the class ‘knife’, that has some resemblance to the objects being detected, does appear in MS COCO. In our problem we consider two classes of firearms: ‘handgun’ and ‘long gun’, respectively. Note that the class ‘handgun’ includes ‘pistol’ and ‘revolver’ object instances, while class ‘long gun’ includes ‘machine gun’, ‘shotgun’ and ‘rifle’ instances.

Table 1 summarizes some important training parameter values used for YOLO model in the experiments. These values were determined under experimentation.

Table 1 Training hyperparameter values used for the YOLOv5 networks.

Network hyperparameter	Set value	
Training epochs	100	
Batch size	32	
Optimizer	SGD	
Learning rate	’lr0 (initial)’: (1, 1e−5, 1e−1)	
	’lrf (final)’: (1, 0.01, 1.0)	
Momentum	(0.3, 0.6, 0.98)	
Decay	(1, 0.0, 0.001)	

All ML algorithms and programs were coded in Python using the OpenCV Computer Vision library and the PyTorch framework for deep learning. These codes and information about the project can be downloaded from: https://github.com/patriciacs99/WeaponDetectionYOLOv5.

Regarding the computing infrastructure (operating system, hardware, etc.), all experiments were performed using the Google Colab environment on a standard Nvidia T4 Tensor GPU provided by Google Colab. The average test detection time per image, using the trained YOLOv5m (medium) model, was 0.5 ms (ms).

Description of the used datasets

In order to carry out our experiments, it was necessary to build our dataset with images containing both classes of objects being detected: ‘handguns’ and ‘long guns’, respectively. The images of ‘handguns’ were taken from the Weapon Detection dataset (Olmos, Tabik & Herrera, 2018) provided by the Data Science and Computational Intelligence (DaSCI) Institute, University of Granada (Spain). As the existing image labelling in this dataset does not match the format required to use our YOLO model, the tool Roboflow has been used to automatically transform the existing XML tags to the new format.

We increased our dataset with images of the class ‘long gun’ taken from free Google Images, and websites such as Depositphotos (2009), Shutterstock (2003) and Pixabay (2023). These images have been properly labeled using Roboflow (Dwyer & Hansen, 2022) in order to complete our dataset, which is summarized in Table 2. Note that the number of images per class is nearly 50%-balanced and, approximately, 70% of these images were used for training, 15% for validation and 15% for tests, respectively. It is interesting to remark that regarding the number of weapon instances per image in the dataset, 91.8% of images (5,441) contain only one firearm, while the remaining images (483) have two or three instances.

Table 2 Image dataset composition and its distribution.

Class	Images	Train	Validation	Test	
handgun	2,972	2,080	446	446	
long gun	2,953	2,052	452	446	
Total	5,924	4,132	898	892	

Additionally, we included some experimentation using the Localization of Firearm Carriers (LFC) dataset (Mahmood et al., 2024). It contains 3,128 images, each depicting at least one human interacting with a firearm in various scenarios. This dataset supports the training and evaluation of machine learning models for firearm detection, aiding security and surveillance systems. It encompasses a range of complexities like varying crowd densities or partially concealed firearms.

Results

This section first describes the detection performance metrics (Padilla et al., 2021) used in the experiments. After that, we show and analyze the results achieved by different experiments, which consider different types of variabilities. The ’medium’ scale YOLOv5m model has been used as reference for this purpose.

Performance metrics

Model evaluation is the process of assessing how well a machine learning model performs on unseen data. To define basic accuracy measures over the detections, it is necessary to consider the following threshold parameters: network confidence loss and IoU (Intersection over Union) thresholds, respectively. Network confidence loss measures the reliability of the network concerning the object class of each computed bounding box. IoU (also called Jaccard Index) measures how accurately an object is detected within a test image. It is computed as the overlapping area between a predicted detection and its corresponding ground truth, divided by the area of the union between the predicted detection and the ground truth. For multi-class detection problems, the mean IoU for an image is calculated by taking the IoU of each class and averaging them. This can be extended to all the images of the test dataset to have an average IoU value. A confidence threshold Confth is used to determine if a network gives a positive answer relative to a detected object in the image. An IoU threshold IoUth is used to determine whether the overlapping between network detection and the ground truth is significant or not. In our framework, the values of these parameters were set to Confth = 0.5 and IoUth = 0.45, respectively.

In our context of firearm detection, we define the true positives (TP), false positives (FP), true negatives (TN) and false negatives (FN), in relation to the detection produced on the images, as follows. Let Conf(p) be the confidence loss returned by the network on the detection of the firearm p present in image i, and IoU(p) the intersection over union value for the same firearm, then p is considered as a TP, FP, TN or FP when any of the following conditions holds: (1) TPp=Confp>=ConfthANDIoUp>=IoUth

(2) FPp=Confp>=ConfthANDIoUp<IoUth

(3) TNp=Confp<ConfthANDIoUp<IoUth

(4) FNp=NPi−|TPi|

where in the last (FN) formula, NP(i) and |TP(i)| represent respectively the number of firearms present in the image i and the number of TP in the same image, respectively. We also accumulate the numbers of TP, FP and FN detections for each image i, and also for the whole dataset to present our test results for YOLOv5. For simplicity, we also denote these accumulated values of TP, FP and FN in the whole test dataset in this form.

Using these accumulated values, some metrics such as precision, recall, and F1-score can be computed as follows: (5) Precision=TPTP+FP

(6) Recall=TPTP+FN

(7) F1−score=2Precision×RecallPrecision+Recall.

Precision represents the fraction of relevant instances among the retrieved instances (i.e., measure of quality), while recall represents the fraction of relevant instances that were retrieved (i.e., measure of quantity). F1-score is defined as the harmonic mean of precision and recall.

Another considered metric is the average precision (AP), that summarizes the Precision-Recall curve, computed as the weighted mean of precisions achieved at each threshold, with the increase in recall from the previous threshold used as the weight: (8) AP= ∑nRecalln−Recalln−1Precisionn

where Precisionn and Recalln are respectively the precision and recall values at the n-th threshold. The mean Average Precision (mAP) is computed as the average of APs for each considered class in the problem, as follows: (9) mAP=1N∑i=1NAPi

where N is the number of classes. In our problem, we have two classes, referred as ‘handgun’ and ‘long gun’, respectively. In some of our experiments, we show the values of mAP50 (i.e., mAP calculated at IOU threshold 0.5) and mAP50 − 95 (i.e., average mAP over different IoU thresholds, from 0.5 to 0.95, with a step of 0.05).

Experimental results

Next, we describe each of the experiments and summarize the corresponding results. First, we present an initial experiment, used as baseline, where global detection results of firearms are presented (Experiment 1). Next, the following experiments using our dataset (Experiments 2 to 6) respectively correspond to each considered individual distortion applied to the test images, namely: impulse noise, occlusions, blurring, image darkening and image shrinking. Finally, the last experiment (Experiment 7) presents the achieved detection results using the LFC dataset.

Experiment 1: Global classification of firearms

This first experiment shows the first detection results produced for each of the two classes in the problem (‘handgun’ and ‘long gun’, respectively), as well as globally (i.e., without distinguishing the classes). These results were produced using a “medium” scale YOLOv5 model (referred as YOLOv5m). This version is characterized by the following reported features (Jocher, 2020): 21.2 million parameters, model size of 41 MB (at FP16 half floating-point precision), 8.2 ms inference time per image (on the Nvidia V100 GPU) and 45.2 mAP based on the original MS COCO dataset (Lin et al., 2014). All these values were achieved for default image sizes of 640 × 640.

The model was parameterized and trained as explained in Section ‘Network Parameterization and Training Details’. The outcomes of this first experiment will be considered as the “baseline” solution, and they will be useful to characterize and compare how the different variabilities influence the detection results.

Table 3 illustrates the achieved baseline results using the ‘medium’ YOLOv5m model on the 892 considered test images (see Table 2).

Table 3 Detection results for classes ‘handguns’ and ‘long guns’, and also for all the firearms using the medium YOLOv5m model.

Class	Instances	Precision	Recall	F1-score	mAP50	mAP50-95	
handgun	522	0.917	0.741	0.820	0.842	0.577	
long gun	502	0.994	0.922	0.957	0.959	0.883	
All	1,024	0.955	0.832	0.890	0.900	0.730	

First, we observe that for both classes (balanced in the number of test instances), the Precision is very high (above 90%) meaning a very low number of FP in the detections. However, although the Recall values are high for the two classes (above 74%), this metric is considerably higher for ‘long gun’ class. It means around a 20% more FN in the class ‘handgun’, which is probably caused by the smaller size and high variability of these weapons within the test images. Consequently, and due to the Recall differences between the classes, the value of F1-score is a 14% higher for the ‘long gun’ class.

The higher mAP for IoU threshold value of 0.5 (mAP50) determines that the YOLOv5m model is very accurate with respect to the detection of both classes (although it performs better for ‘long gun’), since mAP compares the ground-truth bounding boxes to the corresponding detected boxes and returns a score. In the case of mAP50-95, a sequence of IoU threshold between 0.5 and 0.95 the mAP was computed. The obtained metric is around a 37% higher for ‘long guns’ when compared to ‘handguns’, meaning that the detection of ‘long guns’ objects is more robust and stable during inference time.

Now, in this same experiment we use a “small” scale YOLOv5 model (called YOLOv5s), in order to determine how the model scale influences the results. YOLOv5 provides five scaled versions: YOLOv5n (nano), YOLOv5s (small), YOLOv5m (medium), YOLOv5l (large), and YOLOv5x (extra large), where the width and depth of the convolution modules vary to suit specific applications and hardware requirements.

The YOLOv5s version is characterized by the following reported features (Jocher, 2020): 7.2 million parameters, model size of 14 MB (at FP16 half floating-point precision), 6.4 ms inference time per image (on the Nvidia V100 GPU) and 37.2 mAP based on the original MS COCO dataset, where all these values were achieved for default image sizes of 640 × 640.

Table 4 illustrates the achieved detection results using the ‘small’ YOLOv5s model.

Table 4 Detection results using the small YOLOv5s model.

Class	Instances	Precision	Recall	F1-score	mAP50	mAP50-95	
handgun	522	0.870	0.716	0.785	0.816	0.552	
long gun	502	0.973	0.928	0.949	0.961	0.839	
All	1,024	0.921	0.822	0.870	0.889	0.695	

On a global level, as in the previous experiment, using YOLOv5s the class ‘long gun’ is better detected than ‘handgun’ presenting a 17% higher F1-score. Similarly to the previous experiment, and using the YOLOv5s model, the Precision values are very high for both classes (above 87%), with the Recall being a 21% higher for ‘long gun’. Regarding the mAP metrics, mAP50 and mAP50-95, both are favourable to the class ‘long gun’ with differences of 15% and 34% for mAP50 and mAP50-95, respectively.

Prior to the comparison of this ‘small’ model with the previous ‘medium’ one, as YOLOv5s is nearly three times smaller regarding the number of parameters and memory size, the detection results were expected to be much worse for YOLOv5s. Surprisingly, and for all the considered detection metrics, the results are only slightly worse for YOLOv5s. By analyzing the F1-score for all test instances, we only observe a small difference between both models (0.89 for YOLOv5m vs. 0.87 for YOLOv5s, respectively). Concerning the computed mAP metric values, and after having analyzed them for all test instances (i.e., without separating them into classes), we found that the differences are similar in favour of YOLOv5m: a 1.1% better and a 4.8% better for mAP50 and mAP50-90, respectively.

Therefore, from this experiment we conclude that the size of the analyzed models (YOLOv5s vs.YOLOv5m) has very little influence on the firearm detection results. Consequently, it could also be used for the considered problem a “smaller” YOLO model, such as YOLOv5s, since it yields similar results using less computational resources.

We observe that the results of this first experiment show the same tendency (when comparing YOLOv5s with YOLOv5m) as the ones presented by Salahin et al. (2023) and by Rahaman et al. (2022) for caries lesions and blood cells detection, respectively.

Experiment 2: Influence of the impulse noise level

In the following experiments, we analyze the effect of three types of synthetic image distortions: impulse noise (i.e., salt-and-pepper), occlusions and blur, respectively. These have been added to the test images only. The YOLOv5m model has been trained with firearms images which do not include such distortions, and we want to quantify how the increase of impulse noise will affect the detection performance of the network. We have also taken the results from Experiment 1 as a baseline reference for comparison purposes.

The Augmentor software (Bloice, Stocker & Holzinger, 2017) was used to produce the synthetic salt-and-pepper noise distortions on test images for this experiment. First, we generated three subsets of test images with 0.01, 0.02 and 0.05 noise levels. Each of these subsets contains the same test images as in Experiment 1, which are now, respectively, altered with the aforementioned noise distortions. A salt-and-pepper x-level noise, where 0 ≤ x ≤ 1, means to randomly set a 100⋅x percentage of pixels in the image to completely white or completely black. In some cases, it is possible to speckle noise (uniformly added), only as either white (salt) or black pixels (pepper).

Table 5 displays the produced quantitative detection results separated by impulse noise levels and by classes of firearms. From this table, we observe similar high precision results for both classes (slightly favourable to ‘long gun’ class), which means a relatively low number of FP. However, in the particular case of the recall metric the ‘long gun’ class gets specially favored. In general, the difference in recall between classes increases and the recall value itself drops as the noise gets increased. In particular, from 0.01 to 0.05 impulse noise increasing, the recall difference between ‘long gun’ and ‘handgun’ increases from a 42.9% at a 0.01 noise level to 80.2% at a 0.05 noise level. With this same noise level increasing, the value of recall decreases a 64.6% for the class ‘long gun’ and a 87.7% for the class ‘handgun’, respectively. The observed reduction in recall for both classes for increasing noise is reflected in the corresponding drop of the F1-score metric values.

Table 5 Detection results for firearm classes and different noise levels.

Noise	Class	Instances	Precision	Recall	F1-score	
Noiseless	handgun	522	0.917	0.741	0.820	
Noiseless	long gun	502	0.994	0.922	0.957	
0.01	handgun	522	0.891	0.408	0.560	
0.01	long gun	502	0.932	0.715	0.810	
0.02	handgun	522	0.902	0.213	0.345	
0.02	long gun	502	0.908	0.570	0.700	
0.05	handgun	522	0.867	0.050	0.095	
0.05	long gun	502	0.888	0.253	0.394	

Table 6 shows the detection results separated by noise levels, this time without taking into account the firearm classes. In this case, we observe that the number of FP is very low for all noise levels (although this number decreases as the noise level increases). However, the number of FN (which produces non-detections of weapons) is significantly high for all noise levels. It raises from 176 to 448 (that is to say, a percentage growth of nearly a 155%) from noiseless images to the equivalent ones with a low (0.01) noise increase. As the level of noise rises, the number of FN also increases but to a more reduced rate, around a 39% from 0.01 to 0.02 and also from 0.02 to 0.05, respectively. Consequently, the F1-score metric drops due to the noticeable decrease in recall with growing impulse noise. Regarding mAP50 values, using the IoUth = 0.5, we observe that the detection quality drops as the level of noise increases, but not as abruptly as for the F1-score metric.

Table 6 Instance noise detection results without separating into firearm classes.

Noise	Instances	TP	FP	FN	Precision	Recall	F1-score	mAP50	
Noiseless	1,024	848	73	176	0.955	0.832	0.890	0.900	
0.01	1,024	576	56	448	0.912	0.562	0.680	0.752	
0.02	1,024	401	42	623	0.905	0.391	0.520	0.659	
0.05	1,024	157	22	867	0.877	0.151	0.240	0.520	

With this experiment, we prove that for our weapon detection problem the number of object detections (either correct or erroneous) significantly drops as the level of noise rises. As the network was trained with noiseless images, we conclude that a noise distortion increase might drastically affect its detection performance. This effect is also displayed in Fig. 4, where we illustrate some qualitative detection results for the same test image, which is affected by the three considered synthetic levels of impulse noise. Note that for the 0.01 noise level the two handguns present in this image are correctly detected by YOLOv5m with high confidence values (0.7 and 0.9, respectively). When doubling the level of noise, the weapon previously detected with a lower confidence level is now undetected (i.e., a single FN result). The second weapon is still detected, but this time with a lower confidence result (from 0.9 to 0.81). Finally, for the case of a 0.05 noise level both handguns remain undetected (i.e., producing two FN results).

Figure 4 Effect of salt-and-pepper noise on the same test image containing two handguns: (left) 0.01-level, (center) 0.02-level and (right) 0.05-level.

(Note: The original image within this figure can be found in the DaSCI Weapon Detection dataset, and it is distributed under the terms of the Creative Commons Attribution 4.0 International).

Experiment 3: Influence of the number and the size of occlusions

In this test, we have also included the results of Experiment 1 as a baseline reference for comparison purposes. The Roboflow tool (Dwyer & Hansen, 2022) has been used to produce synthetic occlusions on test images for this experiment. In particular, the Cutout functionality that was introduced for YOLOv4 as a data augmentation technique. The Cutout operation randomly masks out a given number of square image regions by setting them to black. It provides two settings for configuration purposes: percent (i.e., size of the cutout region with respect to the global image) and count (i.e., number of cutouts per image), respectively. In this experiment, we have considered three occlusion variants: percent = 30% and count = 1 (referred in our tables as: 1 × 30%); percent = 30% and count = 2(referred as: 2 × 15%); and percent = 30% and count = 3 (referred as: 3 × 10%), respectively. Note that for each of these considered configurations the total occluded area represents the same percentage of the original image (i.e., 30%).

Of course, there are also partial occlusions of firearms that occur when these are being held by their respective users. Nevertheless, their sizes are very difficult to estimate in images. Consequently, we pretend to simulate the presence of these partial occlusions in a more objective way by using different occlusion sizes. We will also place these occlusions at random positions within the original images, which is expected to affect the detection of the firearms contained in them.

Table 7 shows the produced quantitative detection results separated by considered types of Cutout occlusion and by classes of firearms. From this table, we observe similar high Precision results for both classes (slightly favourable to ‘long gun’ class, as in the previous experiment), which means a relatively low number of FP. Differently to the previous experiment, now the Recall values are also high for both classes (although better for the ‘long gun’ class). As a consequence, the combined F1-score metric will have the same tendency for both types of firearms. In general, the detection results for the two considered classes are much better compared to the previous random noise insertion experiment. Compared to non occluded images, the worst reduction in the F1-score metric corresponds, for both firearm classes, to the 1 × 30% occlusion and it approximately represents a 20% for each class.

Table 7 Detection results for firearm classes and different occlusion degrees.

Degree	Class	Instances	Precision	Recall	F1-score	
No occlusion	handgun	522	0.917	0.741	0.820	
No occlusion	long gun	502	0.994	0.922	0.957	
1 × 30%	handgun	522	0.733	0.618	0.671	
1 × 30%	long gun	502	0.788	0.820	0.804	
2 × 15%	handgun	522	0.759	0.667	0.710	
2 × 15%	long gun	502	0.832	0.858	0.845	
3 × 10%	handgun	522	0.773	0.901	0.832	
3 × 10%	long gun	502	0.849	0.901	0.874	

Next, Table 8 shows the detection results separated by occlusion levels, this time without considering the firearm classes. In this particular case, we observe similar values of FP and FN for the three considered occlusions. These values get slightly worse when only one 30% occlusion patch is present. Compared to the non-occlusion case, the most significant percentage increment corresponds to the number of FP that raises nearly a 230% for the 1 × 30% case. Precision and Recall metrics drop around a 20% and a 14%, respectively when compared to non-occlusion case. In the worst case (1 × 30%), with respect to F1-score, there is only a 17% reduction compared to the case lacking occlusions. Similar results have been obtained for the mAP50 metric.

Table 8 Instance occlusion detection results without separating into firearm classes.

Degree	Instances	TP	FP	FN	Precision	Recall	F1-score	mAP50	
No occl.	1,024	848	73	176	0.955	0.832	0.890	0.900	
1 × 30%	1,024	760	240	264	0.760	0,719	0.740	0.728	
2 × 15%	1,024	791	204	233	0.795	0.763	0.780	0.765	
3 × 10%	1,024	827	193	197	0.811	0.800	0.800	0.795	

From this experiment, we conclude that the effect of random synthetic occlusions on object detection performance is much less severe for the considered problem than the addition of random salt-and-pepper noise to the images. Moreover, from the three occlusion configurations experimented, where all of them together hide the same area of the image, the worst one in detection results is 1 × 30%. This case removes a unique larger portion of the image (and probably a larger portion of the target objects contained on it), which makes more difficult for the YOLOv5m model to locate and classify the involved firearm(s), since it was trained with images that do not contain these occlusions.

Figure 5 illustrates some qualitative detection results for the same test image, which is affected by the three considered random synthetic occlusions: one occlusion representing a 30% of the total image size, two occlusions of 15% size each and three occlusions of 10% size each, respectively. Although the total surface occluded in all these images is the same, the effect on the quality of the corresponding detections (as well as their network detection confidences) is worse in the case of one large occlusion than when using several smaller ones. Note that as the Cutout function of Roboflow randomly chooses the position of the occlusions, in some images a large portion of the object is occluded (specially when the selected size of the occlusion is large), while in others the occlusion section is smaller (specially when the selected size of the occlusion is small). Lastly, it is timely to say that some images are not even occluded at all.

Figure 5 Effect of synthetic random occlusions on the same test image: (left) one occlusion of 30% image size, (center) two occlusions of 15% and (right) three occlusions of 30%.

(Note: The original image within this figure can be found in the DaSCI Weapon Detection dataset, and it is distributed under the terms of the Creative Commons Attribution 4.0 International).

Experiment 4: Influence of the Gaussian blur

In this experiment, we analyze the effect in firearm detection results when applying a synthetic Gaussian blur (or Gaussian smoothing) to the test images using different kernel sizes.

The blur—image software (Pinetools, 2022) was used to produce the synthetic Gaussian blurred test images for this experiment. First, we generated three subsets of test images with kernel sizes of 3, 9 and 13 (i.e., by gradually increasing the blur levels), respectively. Each of these subsets contains the same test images of Experiment 1, which are now, respectively, altered with the blur distortions.

Table 9 shows the produced quantitative detection results separated by blur levels and by classes of firearms. From this table, we observe similar high Precision results for both classes (slightly favourable to the ‘long gun’ class), which means a relatively low number of FP. However, in the case of recall metric the differences are much higher in favour of the ‘long gun’ class (around 20%). This Recall decreasing for both classes as the noise increases, is reflected in the corresponding fall of F1-score metric values.

Table 9 Detection results for firearm classes and different levels of blur.

Kernel size	Class	Instances	Precision	Recall	F1-score	
No blur	handgun	522	0.917	0.741	0.820	
No blur	long gun	502	0.994	0.922	0.957	
3	handgun	522	0.901	0.713	0.796	
3	long gun	502	0.981	0.916	0.947	
9	handgun	522	0.894	0.661	0.760	
9	long gun	502	0.974	0.886	0.928	
13	handgun	522	0.894	0.659	0.759	
13	long gun	502	0.971	0.871	0.918	

Next, Table 10 shows the detection results separated by blur levels, but this time without taking into account the firearm classes. In this case, we observe that the number of FP is low and pretty similar for all noise levels. However, the number of FN (which produces non-detections of firearms) is significantly higher for all noise levels. Nonetheless, it only raises from 176 to 232 (that is to say, an increase percentage of nearly a 32%) from images without blur to the corresponding ones with the higher level of blur analyzed (13). The respective Precision values are kept very high (i.e., above 0.93 for all the levels of blur considered, while the decreasing in Recall is not significant as the blur grows. Consequently, the F1-score metric, is little affected by blur increase. Regarding mAP50 values with IoUth = 0.5, we observe a behavior similar to the one described for the F1-score metric.

Table 10 Instance blur detection results without separating into firearm classes.

Kernel size	Instances	TP	FP	FN	Precision	Recall	F1-score	mAP50	
No blur	1,024	848	73	176	0.955	0.832	0.890	0.900	
3	1,024	843	53	181	0.941	0,814	0.873	0.890	
9	1,024	802	57	222	0.934	0.774	0.847	0.867	
13	1,024	792	58	232	0.932	0.765	0.840	0.862	

With this experiment, we show that the number of object detections (either correct or erroneous) is little affected by blur increase in the context of this particular problem. As the network was trained with images without blur, we conclude that this distortion, along with its successive increments, has little impact on the network detection performance.

This effect is also shown in Fig. 6, where we applied the three considered levels of blur to the very same test image. Even though we observe that the “handgun” class in the test image remains detected, the confidence level in the detection decreases by 20% as the Gaussian kernel size varies from k = 3 to k = 13.

Figure 6 Effect of Gaussian blur using tested kernel sizes k on the same test image: (left) k=3, (center) k=9 and (right) k=13.

(Note: The original image within this figure can be found in the DaSCI Weapon Detection dataset, and it is distributed under the terms of the Creative Commons Attribution 4.0 International).

Experiment 5: Influence of image darkening

In this experiment, we analyze the effect in firearm detection results when applying different levels of darkening on the images. This transformation, which involves reducing the brightness or exposure of the images, simulates low ambient light conditions that are typical in surveillance scenarios or medical imaging. The darkening distortion has been implemented using a simple gamma correction to all image pixels using the formula: (10) Ii,j′=c⋅Ii,jγ

where: Ii,j and Ii,j′, respectively, represent all pixels (i, j) of the original and transformed images, c is a positive constant (in our case, c =1), and γ is a positive power used to compensate the nonlinear response of display systems (like monitors) and the human visual system. The γ values can theoretically range from very small positive values (0 < γ < 1) in order to make an image clearer, to large values (γ > 1) in order to darken an image. In our experimental setup, we tested three increasing γ values (1.5, 3 and 5, respectively) in order to determine how this parameter will affect the firearm detection results.

Table 11 shows the achieved quantitative detection results separated by the considered γ darkening levels and by the classes of firearms. From this table, as in previous experiment, we observe similar high precision results for both classes (slightly favourable to the ‘long gun’ class), which means a relatively low number of FP. However, in the case of recall metric the differences are much higher in favour of the ‘long gun’ class (around 21%) for all γ values, and there is also a significant drop for γ > 3. This recall decreasing for both classes as the darkening increases, is reflected in the corresponding fall of F1-score metric values.

Table 11 Detection results for firearm classes and different levels of darkening (gamma correction).

Gamma correction	Class	Instances	Precision	Recall	F1-score	
No darkening	handgun	522	0.917	0.741	0.820	
No darkening	long gun	502	0.994	0.922	0.957	
1.5	handgun	522	0.912	0.716	0.802	
1.5	long gun	502	0.987	0.910	0.947	
3	handgun	522	0.887	0.630	0.737	
3	long gun	502	0.976	0.807	0.883	
5	handgun	522	0.841	0.506	0.632	
5	long gun	502	0.959	0.604	0.741	

Next, Table 12 shows the detection results separated by γ levels but without considering firearm classes. In this case, we observe that the number of FP is low and relatively similar for all darkening levels (being minimal for γ = 1.5). However, the number of FN (which produces non-detections of firearms) is significantly higher and increases as γ does. Respective precision values are kept very high (i.e., above 0.90 for all the levels of darkening considered, while the decreasing in recall is significant, especially when γ =5 that drops to 0.555. Consequently, the F1-score metric, is more affected as darkening increases (note that when γ = 5, the F1-score value decreases by a 23% when compared to the original non-darkened image). Regarding mAP50 values with IoUth = 0.5, we observe a better behavior to the one exhibited by the F1-score metric (i.e., a 18% of worsening).

Table 12 Instance darkening results without separating into firearm classes.

Gamma correction	Instances	TP	FP	FN	Precision	Recall	F1-score	mAP50	
No darkening	1,024	848	73	176	0.955	0.832	0.890	0.900	
1.5	1,024	833	44	191	0.950	0,813	0.876	0.889	
3	1,024	736	55	288	0.931	0.719	0.811	0.837	
5	1,024	568	63	456	0.900	0.555	0.687	0.739	

With this experiment, we show that the number of object detections is relatively affected as the darkening increases (respectively, around 22% of F1-score decrease for both classes between no darkening and highest γ tested). As the network was trained with images without γ correction increments, we conclude that this distortion, along with its successive increments, has a mid-range impact on the network detection performance.

This effect is also shown in Fig. 7, where we applied the three increasing γ levels considered (γ = 1.5, 3 and 5, respectively) to darken the same test image. We can notice that the network misclassifies the type of firearm detection when the γ value doubles from from 1.5 to 3. Note also that for the case of γ equal to 5 the long gun present in the image is not detected.

Figure 7 Effect of image darkening using different gamma corrections on the same test image: (left) γ = 1.5, (center) γ=3 and (right) γ=5.

(Note: The original image within this figure comes from Wikimedia Commons, and it is distributed under the terms of the Creative Commons Attribution 3.0 Unported license).

Experiment 6: Influence of image shrinking

This experiment analyzes the effect in firearm detection when applying different shrinking factors to the images. This transformation (also known as downsampling) that reduces the spatial image resolution by a factor, which simulates long distances between the camera and the target objects that is useful for practical applications like surveillance or industrial automation. In our approach, shrinking has been implemented by first reducing the image spatial resolution by 2 in rows and columns (i.e., the image contains one quarter of its original pixels), and then we apply an upsampling on the “reduced” image using different interpolation algorithms to recover its original spatial resolution. In particular, we apply the following pixel interpolation methods (Jakhetiya, Kumar & Tiwari, 2010): nearest neighbor (the simplest and fastest, assigning the value of the nearest pixel to new pixels, but resulting in blocky images), bilinear (that uses a 2 × 2 pixel neighborhood for smoother results but lacks fine detail), bicubic (that uses a 4 × 4 neighborhood and offers better smoothness and detail) and Lanczos (that employs larger neighborhoods and the sinc function for providing the highest quality though it is the most computationally intensive), respectively.

Table 13 shows the quantitative detection results achieved by the considered interpolation methods applied and the classes of firearms. From this table, we can observe a similar high precision results for both classes (slightly favourable to the ‘long gun’ class), despite the interpolation method used, which means a very low number of FP. However, regarding the recall metric we observe that, as expected, the worst results were achieved using the nearest neighbor interpolation which produces a worsening of around 44% for the “long gun” class and 42% for the “handgun” class, respectively. For the remaining interpolation methods the results are better according to their computational complexity. The recall effect for both classes is reflected in the corresponding decreasing of F1-score values.

Table 13 Detection results for firearm classes and different levels of shrinking.

Interpolation	Class	Instances	Precision	Recall	F1-score	
No shrinking	handgun	522	0.917	0.741	0.820	
No shrinking	long gun	502	0.994	0.922	0.957	
nearest	handgun	522	0.893	0.418	0.569	
nearest	long gun	502	0.943	0.530	0.679	
bilinear	handgun	522	0.923	0.669	0.776	
bilinear	long gun	502	0.948	0.843	0.892	
bicubic	handgun	522	0.924	0.672	0.778	
bicubic	long gun	502	0.943	0.827	0.881	
Lanczos	handgun	522	0.918	0.642	0.756	
Lanczos	long gun	502	0.935	0.809	0.867	

Next, Table 14 shows the detection results for the different interpolation algorithms used but without distinguishing between the two firearm classes. In this particular case, we observe low values of FP for the different types of interpolation. However, the total number of FN is substantially increased by the complexity of the considered algorithms, varying from a 42% for bilinear to a 306% for the case nearest neighbor. Note that the increments produced by bilinear and Lanczos are quite similar to those of bilinear interpolation. By taking as reference the nearest neighbor interpolation, precision metrics only drops by a 4%, while Recall drops ten times more (43%), respectively, when compared to no-shrinking case. In the worst case (nearest neighbor), with respect to F1-score and mAP50 metrics, there are respective reductions of 30% and 23% when compared to the case of no-shrinking.

Table 14 Instance shrinking results without separating into firearm classes.

Interpolation	Instances	TP	FP	FN	Precision	Recall	F1-score	mAP50	
No shrinking	1,024	848	73	176	0.955	0.832	0.890	0.900	
nearest	1,024	485	43	539	0.918	0,474	0.625	0.707	
bilinear	1,024	774	53	250	0.936	0.756	0.836	0.859	
bicubic	1,024	768	55	256	0.933	0.750	0.832	0.854	
Lanczos	1,024	742	58	282	0.927	0.725	0.814	0.840	

This effect is also shown in Fig. 8, where we applied (from left to right) the respective nearest neighbor, bilinear and bicubic interpolation algorithms as part of the image shrinking distortions. We can notice that the network misclassifies the weapon present in the image when using the nearest neighbor interpolation. Also, in this example, using both the bicubic and bilinear algorithms the network correctly classifies the type of firearm. However, using the bicubic algorithm the network confidence detection only increases 6% when compared to the bilinear one (i.e., from 0.81 to 0.86).

Figure 8 Effect of image shrinking using different interpolation algorithms on the same test image: (left) nearest neighbor, (center) bilinear and (right) bicubic.

(Note: The original image within this figure comes from Wikimedia Commons, and it is distributed under the terms of a Public Domain license).

Experiment 7: Image shrinking results on the new Localization of Firearm Carriers dataset

We performed some experimentation using an additional dataset: the Localization of Firearm Carriers (LFC) standard dataset (Mahmood et al., 2024). In particular, we have chosen the imageshrinking transformation as this is a quite realistic effect that simulates long distances between camera and targets, and it special interest for surveillance applications. The LFC dataset consists of 3,128 images, each depicting at least one human interacting with a firearm in various scenarios. For our experiments, we have only considered the ground truth labels corresponding to the firearms (and not the corresponding ones to the carriers). We performed the same tests as in Experiment 6 and now we randomly choose 100 test images (50 from “handgun” and 50 from “long gun” classed that contained a total 104 firearm instances) from the LFC dataset.

Tables 15 and 16 respectively show the quantitative detection results by the considered interpolation methods applied with and without considering the classes of firearms. Note that these achieved results for the image shrinking distortion on LFC dataset are worse when compared to those achieved using our firearm dataset. This is, in general, due to the fact that the original images on LFC dataset present much darker backgrounds when compared to those ones with which we have trained the YOLO models. After applying the shrinking transformation to the test images, the targets become much more difficult to detect (even for humans). Note that the corresponding detection capabilities of the interpolation algorithms used are the same for our dataset and for LFC. The use of a cross-dataset evaluation (i.e., training with one dataset and testing with another different dataset) provides a better generalizability and robustness for the achieved results.

Table 15 Detection results for shrinking, when separating into firearm classes, on the LFC dataset.

Interpolation	Class	Instances	Precision	Recall	F1-score	
No shrinking	handgun	54	0.804	0.759	0.781	
No shrinking	long gun	50	1.000	0.380	0.551	
nearest	handgun	54	0.899	0.296	0.445	
nearest	long gun	50	0.000	0.000	0.000	
bilinear	handgun	54	0.863	0.667	0.752	
bilinear	long gun	50	0.978	0.400	0.568	
bicubic	handgun	54	0.878	0.667	0.758	
bicubic	long gun	50	1.000	0.500	0.667	
Lanczos	handgun	54	0.875	0.648	0.745	
Lanczos	long gun	50	1.000	0.420	0.592	

Table 16 Detection results for shrinking, without separating into firearm classes, on the LFC dataset.

Interpolation	Instances	TP	FP	FN	Precision	Recall	F1-score	mAP50	
No shrinking	104	59	6	45	0.902	0.570	0.699	0.765	
nearest	104	15	19	89	0.450	0,148	0.223	0.292	
bilinear	104	55	5	49	0.921	0.533	0.675	0.739	
bicubic	104	61	4	43	0.939	0.583	0.719	0.770	
Lanczos	104	56	4	48	0.938	0.534	0.681	0.742	

Discussion

A primary objective of this study was to focus on the viability of YOLO when applied to the firearm detection problem (i.e., localization and specific classification). As usual, this network was pretrained on a large dataset (MS COCO) and then adapted to our smaller firearm training database using transfer learning. To enrich our training dataset, we created five synthetic images from each original one as explained in Section ‘Image Preprocessing and Data Augmentation’. The considered ‘medium’ YOLOv5m model has demonstrated to be very effective for the considered problem, with respective average F1-score and mAP50 metric values of 0.89 and 0.9.

Due to the differences in detection models, datasets and analyzed variabilities, it was not possible to fairly compare our results with those presented by other authors. To compensate for this deficiency we have included additional experimentation using the LFC dataset. Most images in our dataset are of reasonably good quality. In order to transfer our solution to real operating environments and determine its feasibility in them, it is necessary to analyze the effect of these multiple variabilities that hinder the detection of the objects of interest (i.e., firearms). As pointed out, in this study we aimed to analyze the impact of each considered individual artificial distortions (i.e., impulse noise, occlusion, blur, darkening and shrinking, respectively) on the detection results. For such purpose, Experiments 2, 3, 4, 5, 6 and 7 were conducted to determine that the most influential factor to degrade the detection performance was by far the incremental addition of impulse noise. Next, image shrinking, synthetic occlusions and image darkening produced an intermediate degradation in the detection performance. Increasing levels of Gaussian blur produced the lowest impact on the network detection performance amongst all considered distortions. To corroborate this, Table 17 shows the worst average value (i.e., without considering the classes) of F1-score and mAP50 performance degradation for each type of artificial distortion considered in the specific experiments.

Table 17 Worst F1-score and mAP50 average performance degradation for the corresponding “worst” variability considered in each of the experiments.

Variability	Category	No. Experiment	F1-score Degradation (%)	mAP50 Degradation (%)	
Impulse noise	0.05	2	270.8	73.1	
Shrinking	nearest	6	29.8	21.4	
Occlusion	1 × 30%	3	20.3	23.6	
Darkening	5	5	22.8	19.9	
Blur	k = 13	4	5.6	4.2	

Although the YOLOv5 network has been able to successfully detect most of the test firearms in images, there were some limiting cases where the weapons were not detected, wrongly detected or misclassified. We illustrate in Fig. 9 two examples of incorrect detections. On the left image, an object that is a firearm has not been detected as a ‘handgun’ (i.e., producing a FN), and on the right image a detected weapon has been wrongly classified as ‘long gun’ instead of ‘handgun’. Note that this happens more frequently under the presence of a high degree of impulse noise within the images.

Figure 9 Two sample examples of incorrectly detected firearms: (left) false positive (FP) detection and (right) wrongly-classified firearm.

(Note: The original image within this figure can be found in the DaSCI Weapon Detection dataset, and it is distributed under the terms of the Creative Commons Attribution 4.0 International).

Conclusions

This work described an experimental study on the individual impact of some considered artificial image distortions (namely: impulse noise, occlusions, Gaussian blur, image darkening and image shrinking, respectively) on the firearm detection problem. For such purpose, we have established a “reference” detector (i.e., the ‘medium’ scale YOLOv5 architecture, referred as YOLOv5m) and two datasets that include annotated firearm images (classified respectively as ‘handguns’ and ‘long guns’) from different sources.

First, we show that YOLOv5 is a highly effective architecture for the considered detection problem. Relative to the distortions analyzed, the one that significantly worsens the detection performance is the incremental synthetic addition of salt-and-pepper impulse noise to the test images, when the model was trained with noiseless images. Image shrinking, occlusions and image darkening have a moderate impact on the firearm detection performance. For the case of Gaussian blur addition, the impact is not significant.

As future work, we plan to perform new experiments which include other types of artificial distortions on the test images. For example, those ones produced by out-of-focus and motion-blurred conditions. Another interesting analysis is to quantify the detection impact of several combined variabilities (e.g., impulse noise with occlusions) by creating test images containing several types of distortions. Finally, we also aim to propose firearm-specific modifications in the used YOLO architecture model in order to improve accuracy in detections.

Additional Information and Declarations

Competing Interests

Author Contributions

Data Availability

The authors declare there are no competing interests.

Patricia Corral-Sanz performed the experiments, performed the computation work, authored or reviewed drafts of the article, and approved the final draft.

Alvaro Barreiro-Garrido analyzed the data, authored or reviewed drafts of the article, and approved the final draft.

A. Belen Moreno conceived and designed the experiments, analyzed the data, prepared figures and/or tables, and approved the final draft.

Angel Sanchez conceived and designed the experiments, performed the experiments, analyzed the data, performed the computation work, prepared figures and/or tables, authored or reviewed drafts of the article, and approved the final draft.

The following information was supplied regarding data availability:

The code is available at Zenodo: Patricia Corral Sanz. (2024). alvarobarreiro/WeaponDetectionYOLOv5: august2024 (v1.0.0). Zenodo. https://doi.org/10.5281/zenodo.13227907.

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
