# Peer review of "On the influence of artificially distorted images in firearm detection performance using deep learning"

_PeerJ Computer Science, doi:10.7717/peerj-cs.2381_

## Round 0.1 · original submission · Major Revisions

Dear Authors,
Your paper has been revised. In view of the reviewers' reports, it needs major revision before being considered for publication in PEERJ Computer Science.

More precisely, the following issues must be solved:

1) Although, in general, the paper is well-written, it is a bit too long because of material that is not important or can be found elsewhere: more precisely, i) the description of Yolo in Section 2.1 is too long (although Figure 1 is ok), ii) Figure 4 is not significant and can be omitted, iii) The title of section 2.2 should be changed, iv) the caption of Figure 2 should be changed. v) Some expressions are incorrect in English and must be corrected: "new bit faster", "also difficult the detection task", "carious" (several instances), "The choosing of," "codes", "in which regards" (several instances), "A initial"


2) The authors must add two other image transformations that are somewhat realistic: 1) darkening and 2) shrinking the images.
3) impulse noise and blurring are not problems that we typically have with modern cameras, so the experiments must be expanded as requested in order to make the paper useful to researchers and practitioners working on this problem.

4) What types of noise will mainly degrade firearms detection and will not affect other objects? How can we mitigate the degradation of performance? How do the proposed noise models affect SOTA firearms detection methods and some in the references? Standard datasets should be used for comparison. What changes in YOLO will improve firearms detection?

·

Basic reporting

1. The authors have experimentally shown that the performance of YOLOv5 significantly degrades for firearms detection, if the test images are synthetically added impulse noise, blurring, occlusion. I would suggest it is an already known fact that adding a noise in the test data (that was not present in the training data) would degrade the performance. These results are already well known and true for all datasets and objects. There was no need to use firearms as objects to show this result.

2. Authors have done a quite poor survey of firearm detection because some very relevant papers are missing in literature review, datasets, and SOTA comparisons. For example, the papers [1],[4] provide extensive firearm detection datasets, literature review and techniques. The papers [2],[3] not only detect the firearms but also localise the carriers of the firearms in crowds. Authors need to go through these papers and improve their work in the presence of these works.

[1] Detection and Localization of Firearm Carriers in Complex Scenes for Improved Safety Measures in
IEEE Transactions on Computational Social Systems, 2023.
[2] Leveraging orientation for weakly supervised object detection with application to firearm localization in Neurocomputing, 2021.
[3] Localizing firearm carriers by identifying human-object pairs, IEEE International Conference on Image Processing (ICIP).
[4] Orientation Aware Object Detection with Application to Firearms in arXiv:1904.10032v1, 2019.

Experimental design

[1] Already available benchmark datasets by the reference [1], [4] and [2], [3] should be used for experiments. Comparisons with these techniques [1]-[4] is also required. These papers are mentioned in the above response: "Basic Reporting".
[2] Authors need to provide solution to the unseen noise added to the test data. Just reporting that performance degrades is not enough contribution.
[3] YOLO is an existing architecture. Authors need to propose fire-arms specific modifications in this architecture to add contributions to their work.

Validity of the findings

The findings are generic and well known. These findings would hold for any type of imagery. The research question is which types of noise will particularly degrade firearms detection and will not effect other objects? How to mitigate the degradation of performance? How SOTA firearms detection methods [1-4] and some in the references are effected by the proposed noise models? Standard datasets should be used for comparison. What changes in YOLO will improve firearms detection?

Additional comments

Almost all aspects are covered.

Reviewer 2 ·

Basic reporting

The paper describes experimental work showing the effect of three image degradation transformations (impulse noise, Gaussian blur and occlusions) on a Yolo-based weapon detector. In general the paper is well-written, but it is a bit too long because of material that is not important or can be found elsewhere: 1) the description of Yolo in Section 2.1 is too long (although Figure 1 is ok), 2) Figure 4 is not significant and can be omitted.

The title of section 2.2 shoudl be changed as the authors are not actually proposing a "solution" but an experimental setup, or experimental methods. Likewise the caption of Figure 2 should be changed as the paper is not proposing a method.

Some expressions are incorrect in English and must be corrected: "new bit faster", "also difficult the detection task", "carious" (several instances), "The choosing of", "codes", "in which regards" (several instances), "A initial"

Experimental design

The experimental design is in general ok. However, the image transformations are not realistic, except perhaps occlusions. The authors must add other two image transformations that are somewhat realistic: 1) darkening, and 2) shrinking the images. The first effect (which is straightforward to implement) somehow simulates low ambient light (typical in surveillance scenarios). The second effect simulates long distances between camera and subjects (also the case in surveillance). These effects were also used in the experiments of reference (Ruiz-Santaquiteria et al, 2023), perhaps you can also add a few lines commenting this within section 1.1.

Validity of the findings

The results are valid and in line with expectations. However, impulse noise and blurring are not problems that we typically have with modern cameras, so the experiments must be expanded as requested in order to make the paper useful to researchers and practitioners working on this problem.

---

## Round 0.2 · accepted · Accept

Dear Authors,

Your paper has been re-reviewed and it has been accepted for publication in PeerJ Computer Science. Thank you for your fine contribution.

Reviewer 2 ·

Basic reporting

The authors have successfully addressed my concerns.

Experimental design

The authors have successfully addressed my concerns.

Validity of the findings

The authors have successfully addressed my concerns.